# Inter- and intra-microcycle external load analysis in female professional soccer players: A playing position approach

Elba Diaz-Seradilla[1], Alejandro Rodríguez-Fernández[2]*, José Antonio Rodríguez-Marroyo[2], Daniel Castillo[3], Javier Raya-González[3], José Gerardo Villa Vicente[1,4]

1 Faculty of Sciences of Physical Activity and Sports, University of Leon, León, Spain, 2 Faculty of Sciences of Physical Activity and Sports, VALFIS Research Group, Institute of Biomedicine (IBIOMED), University of Leon, León, Spain, 3 Faculty of Health Sciences, Universidad Isabel I, Burgos, Spain, 4 Faculty of Sciences Education, University of Valladolid, Soria, Spain

☯ These authors contributed equally to this work.
* alrof@unileon.es

**Data Availability Statement:** All relevant data are within the paper and its Supporting Information files.

## Abstract

This study analyzes the inter- and intra-differences in external load across the microcycle in professional female soccer players. External load during four consecutive microcycles (i.e., M1, M2, M3, and M4) and training sessions (i.e., MD-4, MD-3, MD-2, and MD-1) and a match day (i.e., MD) were registered in seventeen female professional soccer players (age: 26.3 ± 4.6 years; height: 166.3 ± 6.1 cm; body mass: 59.8 ± 6.8 kg; and body mass index: 21.6 ± 1.7 kg·m$^{-2}$) who belonged to the same team in Spanish first division. A 10-Hz GPS that integrated a 100-Hz triaxial accelerometer was used to register external load. The results showed lower decelerations in M2 compared to M1 and M3 ($p < 0.05$), lower high-intensity distance (>16.0 km·h$^{-1}$) in M3 vs. M2, and greater relative sprint distance (>21.0 km·min$^{-1}$) in M4 vs. M1 and M3 ($p < 0.05$). MD-3 registered the highest load for all variables ($p < 0.05$). Forwards (FWs) performed ($p < 0.05$) significantly more sprints (meters and number > 21.0 km·h$^{-1}$) than central midfielders (CMs) and central defenders (CDs) in MD-2 and MD. Both, fitness and conditioning staff should pay special attention to the external loads for each playing position in training sessions to optimize the training process.

## Introduction

The analysis of external load encountered by female soccer players throughout match play has gained great attention in recent years [1–3]. Previous research reported that elite female soccer players cover around 10000 m during a match [4,5], of which almost 300 m are covered sprinting (20.1–32.0 km·h$^{-1}$) [3]. These athletes perform close to 125 high-intensity actions, with an average duration of 2.3 s each, during a match [6]. Likewise, the literature shows that match play is the most demanding session of the microcycle [7]. As such, strength and conditioning staff could consider this aspect when planning a team's training strategies. In addition, coaches should attend to the quantification of external loads imposed by the weekly training

**Funding:** The author(s) received no specific funding for this work.

**Competing interests:** The authors have declared that no competing interests exist.

microcycle to understand the dose-response nature of the training process and optimize player performance [8].

One of the main goals of coaching staff in elite soccer is to learn the best training periodization to improve players' physical fitness and reduce injury risk, with the intent of optimizing the team's performance [9]. In this sense, some periodization models have been used [10], most of which coincide with proposed high load reduction during the session prior to a competition [11], confirming the concept of tapering [12]. The typical microcycle, when playing a match once a week, is composed of strength, endurance, and speed acquisition training sessions located in the central days of the microcycle in male [13] and female [2] professional soccer players. This distribution across the week allows players to follow the horizontal alternation principle, which consists of maximizing a particular physical capacity while the others recover [14]. As such, lower external training load (total distance and high-speed distance) in comparison to the match day load has been reported in professional male [15] and female [16] soccer players. External load on acquisition microcycle days is characterized by higher distance covered (total and relative) and time > 90% of maximal heart rate on endurance days and high-sprint and high-intensity distance on speed days [14]. Although training periodization is relevant to enhance a team's performance, the organization of the microcycle seems to be a more realistic strategy due to the chaotic nature of women's soccer [15]. Additionally, other authors have demonstrated a higher coefficient of variability in the distances covered at high intensity (20.8%, 26.1%, and 54.5%) and sprinting (65.1%, 94.2%, and 100.2%) in neuromuscular, endurance, and speed sessions, respectively [17].

Inappropriately training loads can cause overuse injuries [18] and increased injury risk is associated with spikes in workload (i.e., overloading) and low chronic workloads (i.e., underloading) [19] so. Given that quantification of the workload has become an essential aspect in soccer performance [11]. Some external factors, such as sex [20], performance level [9], playing surface, and/or match outcomes [3] could influence the external load encountered by female players during matches. However, playing position is considered the most influential variable on external load [21], mainly in terms of high-intensity actions (e.g., sprints or jumps). In this sense, elite female defenders covered significantly less high-intensity distance (1260 ± 110 m) than midfielders and attackers (1650 ± 110 and 1630 ± 110 km, respectively) [10]. In addition, attackers were able to sprint greater distances compared to defenders (0.52 ± 0.03 km vs. 0.33 ± 0.05 km) [9]. Understanding the external training and match loads in female soccer players according to position would allow coaching staff to establish specific training sessions and an adequate distribution of training content within the microcycle.

As research on the analysis of external training and match loads in female professional soccer players is scarce, the aim of this study was to analyze the inter- or intra-differences in external load across the microcycle and compare the external load considering playing positions among microcycles and between training sessions and a match day in professional female soccer players. We hypothesized that there is similar distribution of external loads between each microcycle, but that differences could exist in each training session and match day external load according to playing position.

## Material and methods

### Study design

An observational and descriptive design was implemented to analyze the inter- and intra-differences in the external load encountered by female professional soccer players during four consecutive microcycles, according to playing position. This study was carried out during the 2019–2020 in-season period when each microcycle was composed of four training sessions

and one official match; thus, 16 training sessions and four matches (i.e., 340 individual observations) were registered. Microcycle 1 was performed in the middle of the first part of the season, 4 weeks after the end of the pre-season and demanding similar workload in all players. Training sessions were conducted on the same playing surface (third generation artificial turf) and at the same time (4:30 P.M.). Matches were played on four pitches with similar dimensions ($100 \times 64$ m) and artificial surfaces, comprising two matches at home and two matches away. Two matches were won and two matches lost. During the training sessions, the researchers did not influence the training exercises.

## Participants

Seventeen female professional soccer players (age: $26.3 \pm 4.6$ years; height: $166.3 \pm 6.1$ cm; body mass: $59.8 \pm 6.8$ kg; and body mass index: $21.6 \pm 1.7$ kg·m$^{-2}$) who belonged to the same team in Spanish first division participated in this study. Because the team played in a 1–5–3–2 formation during the four official matches, players were classified according to their playing position: central backs (CBs, $n = 3$), full backs (FBs, $n = 3$), central midfielders (CMs, $n = 6$), and forwards (FWs, $n = 5$). Goalkeepers were excluded from the subsequent analysis due to their specific role. Players were excluded from further analysis if they did not complete a full competitive match or if they had suffered an injury in the 2 months prior to the investigation. Before beginning the study, subjects were informed of the study's objectives, risks, and benefits, and signed the informed consent form. The study was conducted according to the requirements of the Declaration of Helsinki and was approved by the ethics committee of ***for blinded purposes*** code: 004–2021.

## Procedures

External loads were registered during the 4-week period according to the microcycle number (microcycle 1 = M1; microcycle 2 = M2; microcycle 3 = M3; and microcycle 4 = M4) and training session (i.e., days before match day [MD], MD-4, MD-3, MD-2, MD-1, and MD) [15,22]. The usual distribution through microcycle was the following: MD-4: recovery session and general resistance in gym; MD-3: specific resistance and endurance session (i.e., plyometric drills, strength stations and small and medium sided games); MD-2: speed and tactical approximation (i.e., sprint running and large small sided games); and MD-1: activation session (reaction speed and small sided games) similar distribution of male soccer players [14,17] and development/maintenance of the main three physical capacities. Tactical task (i.e., conditioned games in medium and large spaces simulating competition situations, superiorities and inferiorities) represents 20%, 40% and 40% in MD-4, MD-3 and MD-2 respectively. The external load of the microcycle was analyzed only in those players who played $\geq 60$ min in MD. In all training sessions and match play, the players performed a similar standardized 15–20 min warm-up that included running, dribbling, and specific tasks (drills).

The external loads across the training sessions (n = 16) and matches (n = 4) were registered individually for each player using a 10-Hz GPS that integrated a 100-Hz triaxial accelerometer (WIMU PRO, RealTrack Systems, Almería, Spain). This technology has been previously used in soccer research on activity-demand profiles [23] and reported high levels of validity and reliability [24]. To avoid inter-unit variability, each player wore only their assigned unit, which was inserted into the manufacturer-provided vest that holds the receiver tightly between the scapulae. The units were activated 15 min before the start of each training session and match. Following each training session and match, GPS data were downloaded using the specific software package (WIMU SPRO, Almería, Spain) on a personal computer and exported for further analysis. In the absence of unanimity in the determination of thresholds in female soccer

players [25], total distance (TD; in meters), relative distance (RD; in meters/minute), high-intensity distance (HID; meters and $m \cdot min^{-1} > 16.0 \, km \cdot h^{-1}$), sprint distance (SPD; meters, number, and $m \cdot min^{-1} > 21.0 \, km \cdot h^{-1}$), acceleration (ACC; number and $m \cdot min^{-1}$), deceleration (DCC; number and $m \cdot min^{-1}$), and maximal speed ($km \cdot h^{-1}$) were registered. In addition, player load (PL; $AU \cdot min^{-1}$) was computed. The average number of satellites registering data during the measurements was 9.1±1.0 and horizontal dilution of precision was 0.96.

## Statistical analysis

Results are presented as mean ± standard deviation (SD). After confirming the normal distribution of the data, a repeated measure analysis of variance (ANOVA) was conducted to compare the external load of players among each microcycle and each training session and match play. In addition, one-way ANOVA was used to analyze the external load differences among playing positions (i.e., CB, FB, CM, and FW) in each training session and on each match day. When significant differences were obtained, Bonferroni post hoc tests were used. The coefficient of variation (CV) was quantified to assess the variation within the microcycle, training sessions, and match day. Statistical analysis was conducted using SPSS version 25.0 and the significance level was set at $p < 0.05$.

## Results

The external loads imposed on professional female soccer players during four consecutive microcycles are shown in Table 1. M1 showed significantly ($p < 0.05$) lower ACC and DCC distances compared to the other three microcycles, whereas during M3, significantly fewer meters of HID ($p < 0.05$) were observed than in M2. No significant differences in TD, SPD (meters and number), and PL were observed among microcycles. The CVs for each microcycle were 9.4%, 11.8%, 11.1%, and 7.7% for RD; 17.6%, 30.5%, 21.81%, and 18.9% for HID; 4.2%, 6.4%, 6.4%, and 4.8% for ACC; and 4.2%, 6.9%, 7.2%, and 4.8% for DCC in M1, M2, M3, and M4, respectively.

Analysis of external load of the training sessions and matches demonstrated that training session MD-3 presented a significantly higher load ($p < 0.05$) for all external variables (Table 2). All training sessions showed significantly ($p < 0.05$) less external load than MD except ACC and DCC, which were significantly higher ($p < 0.05$) compared to MD. The CV for each training session is reported in Table 2.

**Table 1. Mean external load of four consecutive microcycles.**

| | TD (m) | RD (m·min⁻¹) | HID (m) | HID (m·min⁻¹) | SPD (m) | SPD (m·min⁻¹) | SPD (n°) | Maximal Velocity (km·h⁻¹) | ACC (m·min⁻¹) | DCC (m·min⁻¹) | PL (m·min⁻¹) |
|---|---|---|---|---|---|---|---|---|---|---|---|
| M1 | 21942 ±4488 | 67.5±7.4 | 498.1 ±390.2 | 6.0±4.5 | 305.2 ±181.1 | 3.7±1.4 *† | 17.8 ±11.5 | 23.2±1.4 | 36.7±1.9 | 36.8±1.5 † | 0.8±0.1 |
| M2 | 24139 ±6044 | 71.5±8.8 | 545.4 ±338.6 | 6.3±3.7 | 454.4 ±247.8 | 5.5±2.7 | 23.2 ±15.0 | 24.8±1.0 | 35.5±2.2 | 35.15±1.8 | 0.8±0.2 |
| M3 | 22414 ±4774 | 67.1±8.2 | 504.2 ±353.8 | 5.8±4.0 | 325.2 ±122.7 | 3.4±2.6 *† | 16.0 ±6.6 | 24.0±2.0 | 36.4±1.7 | 36.8±1.3† | 0.9±0.1 |
| M4 | 24251 ±5027 | 65.1±5.0 | 445.4 ±402.2 | 5.1±4.4 | 426.3 ±217.5 | 4.9±2.4 | 20.1 ±10.4 | 23.7±1.0 | 35.9±1.5 | 35.5±1.5 | 0.8±0.1 |

Data are presented according to microcycle: M1 = Microcicle 1; M2 = Microcicle 2; M3 = Microcicle 3; M4 = Microcicle 4; TD = total distance; HID = high intensity distance; SPD = sprint; ACC = accelerations; DCC = decelerations; PL = player Load.

* = denotes difference from M4

† = denotes difference from M2.

**Table 2. Mean external load of training sessions and match day of four consecutive microcycles.**

| | TD (m) | RD (m·min⁻¹) | HID (m) | HID (m·min⁻¹) | SPD (m) | SPD (m·min⁻¹) | SPD (n°) | Maximal velocity (km·h⁻¹) | ACC (m·min⁻¹) | DCC (m·min⁻¹) | PL (AU·min⁻¹) |
|---|---|---|---|---|---|---|---|---|---|---|---|
| MD-4 | 4817 ± 407*†^ | 57.3 ± 4.7*† | 288.1 ± 139.9 | 3.3 ± 1.6 *† | 52.3 ± 6.2*†^ | 0.6 ± 0.4*†^ | 2.6 ± 1.2*† | 23.4 ± 0.3*†^ | 38.1 ± 1.5*†‡^ | 38.1 ± 0.4*†^ | 0.8 ± 0.1*†^ |
| CV (%) | 5.8 | 7.1 | 47.6 | 48.0 | 107.2 | 83.7 | 39.9 | 12.3 | 7.1 | 7.1 | 6.5 |
| MD-3 | 5952 ± 727† | 75.4 ± 8.5† | 699.4 ± 290.1 | 8.7 ± 3.3 † | 151.7 ± 20.3† | 1.7 ± 0.8 | 7.3 ± 3.7† | 25.0 ± 0.3† | 35.7 ± 2.2† | 35.2 ± 0.5† | 0.9 ± 0.2† |
| CV (%) | 24.1 | 10.9 | 41.5 | 38.8 | 31.5 | 59.9 | 14.9 | 3.6 | 8.3 | 8.6 | 15.2 |
| MD-2 | 4682 ± 221*† | 54.3 ± 3.4*† | 420.6 ± 199.3 | 4.7 ± 2.2*† | 57.0 ± 9.5*† | 0.6 ± 0.2 | 3.4 ± 1.9*† | 23.9 ± 0.5*† | 39.2 ± 1.3*† | 39.2 ± 0.4*† | 0.7 ± 0.1*† |
| CV (%) | 15.3 | 8.1 | 47.4 | 47.0 | 28.8 | 17.8 | 23.2 | 2.6 | 8.1 | 8.1 | 10.2 |
| MD-1 | 4083 ± 414*†‡ | 55.3 ± 4.7*† | 215.6 ± 194.9 | 2.9 ± 2.9*† | 24.8 ± 4.2*†‡ | 0.3 ± 0.3*† | 1.9 ± 1.2*† | 21.9 ± 0.3*†‡ | 37.0 ± 1.5*‡ | 37.0 ± 0.4*†‡ | 0.8 ± 0.1*†‡ |
| CV (%) | 31.9 | 15.5 | 90.4 | 99.8 | 89.8 | 59.1 | 88.2 | 7.3 | 24.7 | 24.7 | 27.1 |
| MD | 9347 ± 1013* | 96.3 ± 8.8* | 1110.5 ± 331.8 | 11.9 ± 3.6 * | 235.1 ± 20.6* | 2.6 ± 1.8 | 13.4 ± 4* | 25.3 ± 0.3* | 31.7 ± 1.6* | 31.7 ± 0.4* | 1.3 ± 0.2* |
| CV (%) | 3.7 | 5.5 | 30.8 | 9.9 | 6.5 | 41.9 | 9.1 | 2.5 | 4.0 | 4.1 | 3.2 |

Data are presented according to days to match: TD = total distance; HID = High intensity distance; SPD = sprint; ACC = accelerations; DCC = decelerations; PL = player Load.

* = denotes difference from MD-3

† = denotes difference from MD

‡ denotes difference from MD-2

^ denotes difference from MD-1.

**Table 3. External load in microcycles according to playing position.**

| | | TD (m) | RD (m·min⁻¹) | HID (m) | HID (m·min⁻¹) | SPD (m) | SPD (m·min⁻¹) | SPD (n°) | Maximal Velocity (km·h⁻¹) | ACC (m·min⁻¹) | DCC (m·min⁻¹) | PL (AU·min⁻¹) |
|---|---|---|---|---|---|---|---|---|---|---|---|---|
| M1 | CB | 22524 ±4696 | 62.9±8.4 | 385.7 ±325.7 | 4.6±3.5 | 254.0 ±221.0 | 2.8±1.9 | 16.0 ±14.5 | 22.7±1.4 | 38.6±1.5 | 38.6±1.1 | 0.7±0.1 |
| | FB | 22496 ±8047 | 65.7±4.1 | 508.5 ±365.2 | 6.2±4.3 | 465.8±67.0 | 5.2±0.4 | 21.0 ±18.4 | 24.2±0.7 | 37.0±0.2 | 37.8±0.8 | 0.8±0.1 |
| | CM | 23555 ±4695 | 68.0±6.0 | 471.9 ±421.0 | 5.7±4.7 | 198.5 ±117.1 | 2.2±1.0 | 11.7 ±6.1 | 22.4±1.0 | 36.2±2.0 | 36.3±1.3 | 0.9±0.2 |
| | FW | 19112 ±2306 | 70.2±9.7 | 618.7 ±388.8 | 7.5±5.4 | 420.9 ±183.1 | 4.7±1.8 | 26.0 ±9.7 | 24.1±1.6 | 36.2±2.0 | 36.1±1.2 | 0.9±0.1 |
| M2 | CB | 24108 ±5980 | 64.0±12.1 | 408.3 ±243.9 | 4.7±2.6 | 379.8±87.8 | 4.2±17 | 22.0 ±8.0 | 24.4±1.5 | 36.5±3.7 | 35.0±2.9 | 0.7±0.1 |
| | FB | 23725 ±8414 | 65.4±2.0 | 521.8 ±295.9 | 5.8±2.7 | 504.0 ±351.9 | 5.6±3.1 | 24.5 ±20.5 | 25.8±0.2 | 35.9±1.2 | 35.4±0.8 | 0.7±0.1 |
| | CM | 22510 ±6778 | 73.3±7.8 | 526.5 ±328.7 | 6.1±3.7 | 336.1 ±115.4 | 3.7±2.3 | 17.6 ±5.5 | 24.5±0.8 | 34.9±2.6 | 35.5±1.8 | 0.9±0.2 |
| | FW | 28250 ±3456 | 77.0±5.8 | 720.9 ±405.8 | 8.3±4.5 | 772.0 ±330.5 | 8.6±4.2 | 33.2 ±26.0 | 25.2±1.0 | 35.7±0.9 | 33.4±3.1 | 0.9±0.2 |
| M3 | CB | 21847 ±5530 | 66.3±7.9 | 474.5 ±379.9 | 5.2±4.1 | 272.6±96.7 | 3.0±1.0 | 15.3 ±4.7 | 25.5±3.4 | 36.8±0.6 | 37.7±1.3 | 0.7±0.0 |
| | FB | 19499 ±1001 | 64.4±2.0 | 489.1 ±430.7 | 5.5±4.7 | 348.0±33.5 | 3.0±0.6 | 18.5 ±2.1 | 23.2±0.5 | 36.5±1.3 | 36.9±2.1 | 0.8±0.0 |
| | CM | 23137 ±5252 | 66.6±3.8 | 499.6 ±376.2 | 5.8±4.2 | 281.1 ±111.5 | 3.1±0.9 | 12.3 ±7.2 | 22.6±0.6 | 36.9±1.5 | 36.7±1.3 | 0.9±0.1 |
| | FW | 22414 ±4774 | 69.2±13.9 | 532.9 ±303.4 | 6.3±3.8 | 400.5 ±153.8 | 4.4±1.6 | 21.7 ±3.6 | 25.0±2.0 | 35.7±2.4 | 36.3±1.3 | 0.9±0.2 |
| M4 | CB | 22491 ±7605 | 60.9±3.8 | 273.5 ±217.3 | 3.3±2.3 | 165.7 ±117.2* | 1.8±0.9* | 8.6 ±5.5* | 23.7±1.3 | 37.4±1.2 | 37.2±0.8 | 0.7±0.1 |
| | FB | 23146 ±4389 | 60.8±0.1 | 495.6 ±395.4 | 5.6±4.3 | 583.2 ±234.7 | 6.6±2.7 | 27.0 ±11.3 | 24.7±0.5 | 36.9±0.95 | 36.5±0.1 | 0.8±0.1 |
| | CM | 2423 ±4555 | 67.0±3.6 | 451.0 ±434.7 | 5.2±4.8 | 380.3 ±155.0 | 4.2±1.2 | 17.7 ±7.0 | 23.3±0.9 | 35.3±1.15 | 35.2±1.3 | 0.9±0.1 |
| | FW | 26143 ±5027 | 66.8±6.7 | 530.1 ±438.1 | 6.2±4.9 | 624.0 ±140.5 | 6.9±1.1 | 29.25 ±9.7 | 24.2±0.7 | 35.2±1.6 | 34.7±1.6 | 0.9±0.1 |

Data are presented according to microcycle and positions: M1 = microcycle 1; M2 = microcycle 2; M3 = microcycle 3; M4 = microcycle 4; TD = total distance;

CB = Central defender; FB = fullback; CM = central midfielder; FW = forwards; RD = relative distance; HID = High intensity distance; SPD = sprint;

ACC = accelerations; DCC = decelerations; PL = player Load.

*Denote differences from FW.

The analysis of external load in the microcycles according to playing position showed that FW covered significantly ($p<0.05$) more distance sprinting and performed more sprints than CB in M4 (Table 3).

Table 4 presents the training load in training sessions and match days according to playing position. FWs performed significantly more sprints (in number and meters; $p < 0.05$) than CMs and CBs in MD-2 and MD, and covered significantly more sprint distance (meters; $p < 0.05$) than CBs in MD-2. Comparing the external training load versus MD, FBs, CMs, and FWs in MD-3 exceeded 50% of match play values in sprint distance (Fig 1). In addition, FWs and FBs in MD-3 reached match maximal velocity. Significantly higher peak velocity ($p < 0.05$) was obtained in MD-3 compared to MD-1 for FBs, CMs, and FWs.

**Table 4. Mean external training load variables according to days to match and playing position.**

| | Position | TD (m) | RD (m·min⁻¹) | HID (m) | HID (m·min⁻¹) | SPD (m) | SPD (m·min⁻¹) | SPD (n°) | Maximal velocity (km·h⁻¹) | ACC (m·min⁻¹) | DCC (m·min⁻¹) | PL (AU·min⁻¹) |
|---|---|---|---|---|---|---|---|---|---|---|---|---|
| MD-4 | CB | 4729 ± 454 | 55.4 ± 5.3 | 226.3 ±72.4* | 2.6±0.8* | 40.2 ± 19.1 | 0.4 ± 0.2 | 1.8 ± 0.7 | 23.2 ± 1.5 | 39.3 ± 0.1 | 39.3 ± 0.1 | 0.68 ± 0.1 |
| | FB | 4748 ± 509 | 54.9 ± 5.8 | 239.1 ±111.9 | 2.8±1.3 | 69.2 ± 14.3 | 0.8 ± 0.2 | 3.4 ± 0.9 | 24.3 ± 0.7 | 38.8 ± 0.1 | 38.8 ± 0.1 | 0.71 ± 0.1 |
| | CM | 4984 ± 318 | 58 ± 3.7 | 262.5 ±117.7* | 3.0±1.3* | 48.3 ±16.6 | 0.5 ± 0.2 | 2.1 ± 0.7 | 23.3 ± 0.5 | 37.2 ± 1.6 | 37.2 ± 1.6 | 0.74 ± 0.1 |
| | FW | 4768 ± 862 | 59.7 ± 5.1 | 387.5 ±168.6 | 4.5±1.9 | 61.1 ± 33.1 | 0.7 ± 0.3 | 3.3 ± 1.5 | 23.9 ± 1.7 | 37.7 ± 1.8 | 37.7 ± 1.8 | 0.86 ± 0.1 |
| MD-3 | CB | 5691 ± 921 | 68.6 ± 9.5 | 491.4 ±170.2* | 5.9±2.0* | 91.4 ± 81.5 | 1.0 ± 0.7 | 5.2 ± 5.2 | 24.1 ± 1.9 | 37.2 ± 1.5 | 36.3 ± 2.2 | 0.72 ± 0.1 |
| | FB | 6241 ± 24 | 75 ± 3.1 | 719.7 ±234.6 | 8.9±3.0 | 216.6 ± 74.7 | 2.4 ± 1.0 | 8.9 ± 0.8 | 26.1 ± 1.3 | 34.3 ± 0.3 | 34.3 ± 0.4 | 0.83 ± 0.1 |
| | CM | 6492 ± 969 | 79.8 ± 9.8 | 696.1 ±284.8 | 8.6±3.3 | 114.3 ± 54.1 | 1.3 ± 0.6 | 5.5 ± 1.8 | 24.2 ± 1.1 | 34.8 ± 3.2 | 34.3 ± 2.8 | 0.92 ± 0.2 |
| | FW | 6172 ± 545 | 76.4 ± 4.7 | 820.2 ±328.3 | 10.2±3.4 | 194.8 ± 66.7 | 2.2 ± 0.9 | 9.9 ± 3.4 | 25.6 ± 1.3 | 35.4 ± 1.5 | 34.9 ± 1.7 | 0.95 ± 0.2 |
| MD-2 | CB | 4908 ± 653 | 55.1 ± 7.1 | 389.9 ±359.8 | 4.3±3.9 | 40.6 ± 6.4 | 0.5 ±0.1 | 2.1 ± 0.7* | 24.5 ± 3.7 | 39.2 ± 1.9 | 39.2 ± 1.9 | 0.54 ± 0.1 |
| | FB | 4671 ± 113 | 51.7 ± 1.2 | 475.5 ±110.0 | 5.3±0.8 | 67.2 ± 5.3 | 0.7 ± 0.1 | 4.6 ± 0.9 | 24.3 ± 0.0 | 39.7 ± 0.5 | 39.7 ± 0.4 | 0.65 ± 0.0 |
| | CM | 4880 ± 334 | 55.7 ± 2.1 | 377.9 ±132.4 | 4.3±1.5 | 37.9 ± 29.8* | 0.4 ± 0.4 * | 2.4 ± 1.4* | 23.1 ± 0.9 | 38.9 ± 1.4 | 38.9 ± 1.4 | 0.68 ± 0.1 |
| | FW | 4753 ± 312 | 54 ± 2.3 | 492.4 ±145.9 | 5.6±1.8 | 89.7 ± 36.2 | 1.0 ±0.5 | 5.7 ± 2.1 | 24.5 ± 0.9 | 38.9 ± 1.6 | 38.6 ± 1.7 | 0.73 ± 0.1 |
| MD-1 | CB | 3933 ± 231 | 54.4 ± 2.9 | 150.0 ±143.2 | 2.2±2.4 | 17.6 ± 7.1 | 0.2 ± 0.3 | 1.5 ± 0.7 | 22.3 ± 0.3 | 37.9 ± 0.4 | 37.9 ± 0.4 | 0.71 ± 0.1 |
| | FB | 3978 ± 364 | 52.8 ± 2.7 | 227.9 ±173.1 | 2.9±1.9 | 27.1 ± 15.5 | 0.3 ± 0 .2 | 1.8 ± 0.8 | 22.1 ± 0.6 | 37.7 ± 0.3 | 37.7 ± 0.3 | 0.69 ± 0.0 |
| | CM | 4181 ± 461 | 57.8 ± 5.9 | 221.4 ±206.3 | 3.1±3.2 | 16.7 ± 12.7 | 0.2 ± 0.2 | 1.2 ± 0.9 | 21.4 ± 0.8 | 36.4 ± 1.9 | 36.8 ± 1.9 | 0.77 ± 0.1 |
| | FW | 4113 ± 675 | 55.2 ± 4.3 | 254.9 ±225.2 | 3.3±3.2 | 36.7 ± 25.7 | 0.4 ± 0.3 | 2.6 ± 1.8 | 21.9 ± 1.7 | 36.1 ± 1.6 | 36.1 ± 1.6 | 0.76 ± 0.1 |
| MD | CB | 8025 ± 1241 | 87.7 ± 11 | 717.5 ±231.4*†‡ | 7.5 ±2.3*†‡ | 168.8 ± 78.4 | 1.9 ± 0.7 | 10.3 ± 5.2* | 25.8 ± 2.0 | 33.3 ± 2.2 | 33.2 ± 2.2 | 1.06 ± 0.1† |
| | FB | 9198 ± 675 | 91.2 ± 0.5 | 1178.4 ±139.1 | 12.3±1.7 | 272.9 ± 34.3 | 3.0 ± 0.5 | 15.2 ± 1.2 | 26.1 ± 1.0 | 31.7 ± 1.1 | 31.7 ± 1.2 | 1.16 ± 0.1 |
| | CM | 9895 ± 527 | 102.5 ± 3.8 | 1303.7 ±284.7 | 13.8±3.2 | 206.9 ± 28.6 | 2.3 ± 0.4 | 10.8 ± 0.9* | 24.6 ± 0.9 | 30.9 ± 1.0 | 30.9 ± 1.0 | 1.41± 0.2 |
| | FW | 9597 ± 1121 | 97.4 ± 8.9 | 1205.4 ±215.9 | 11.9±3.7 | 282.4 ± 78.5 | 3.1 ± 1.0 | 17.5 ± 1.6 | 25.7 ± 1.1 | 31.6 ± 1.4 | 31.6 ± 1.4 | 1.27 ± 0.1 |

Data are presented according to days to match and positions. CB = Central defender; FB = fullback; CM = central midfielder; FW = forwards. TD = total distance;

HID = high intensity distance; SPD = sprint; ACC = accelerations; DCC = decelerations; PL = player Load.

* denote differences from FW

† denote differences from CM

‡ denote differences from FB.

## Discussion

The aim of this study was to analyze the inter- and intra-differences in external load across microcycles and compare the external load among playing positions and microcycles and between training sessions and match days in female professional soccer players. The novel

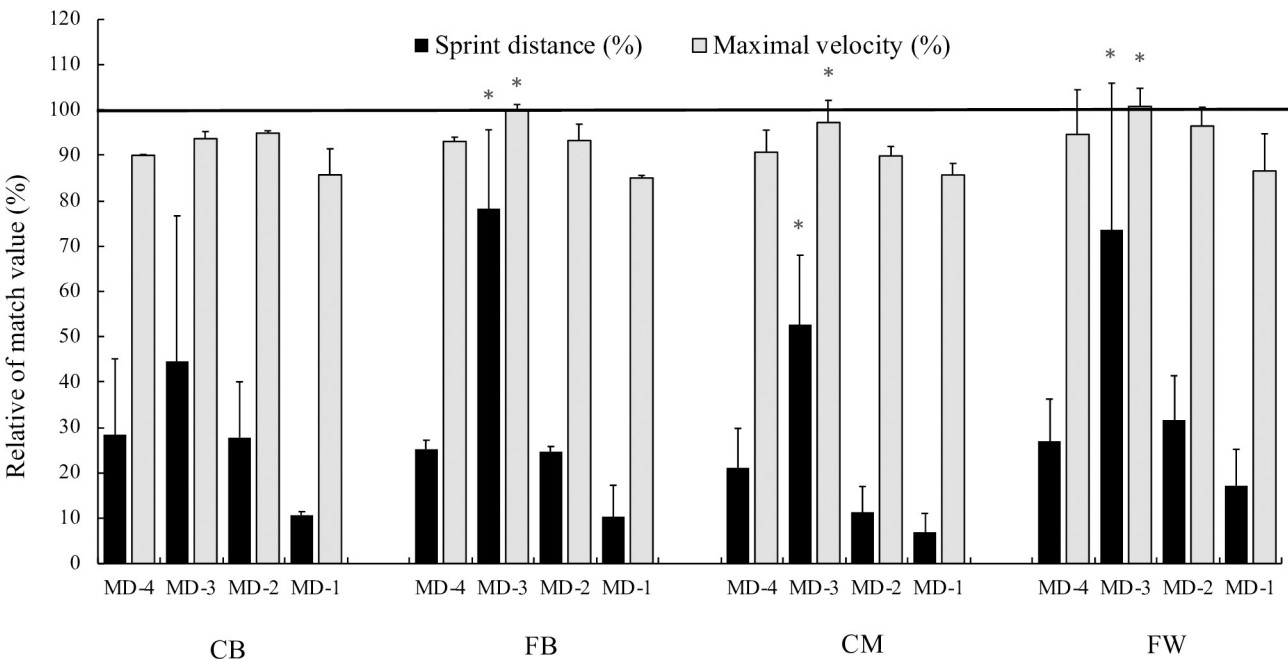

**Fig 1. Sprint distance and maximal velocity for professional players relative to match play in training sessions.** * = denote differences from MD- 1. CB = central defender; FB = fullbacks; CM = central midfielder; FW = forwards.

findings revealed a similar inter-microcycle external training load. In addition, according to playing position, FWs covered significantly (p < 0.05) more sprint distance and performed a greater number of sprints than CBs in M4. Intra-microcycle analysis showed significant (p < 0.05) differences in external load between training sessions, with central sessions of the microcycle demonstrating the highest external load, thus confirming our hypothesis. In addition, during training sessions in MD-3, only FWs and FBs reached maximal velocity comparable to peak match velocity. Our results may aid researchers to understand periodization models and taper strategies, through demonstrating limited training-load-relevant variations across microcycles and that differences between positions only exist in the sprint distance and numbers of sprints.

Quantifying training and match load could facilitate the establishment of specific demand profiles in female soccer players in order to periodize the training sessions as well as the recovery strategies in this specific population. As such, the current study revealed no general inter-microcycle differences in external training loads across the mesocycle, since only M3 and M2 showed a significantly (p<0.05) lower SPD covered in comparison to M4 and M1 (~3.5 m·min$^{-1}$ vs ~5 m·min$^{-1}$). Similarly, the study performed by Owen et al. [22] reported no significant differences in external load across microcycles in male professional soccer players. These findings suggest that external load remains constant in female professional soccer players, in order to maintain their physical capacities to accommodate match demands during the competitive season and reduce injury risk [26]. To prevent injuries, fitness and conditioning staff can manage some external load to determine the correct load for future training sessions while helping to mitigate injuries due to overuse during a long competitive season [26,27].

Another major finding of this study was that external training loads were significantly greatest in the MD-3 sessions compared with the rest of the training sessions, whereas all training sessions showed significantly less external load than observed on match day according to previous studies [28]. The central sessions of the microcycle (i.e., MD-3) produced the greatest

loads in terms of RD, ACC, HID, and SPD, which is supported by the types of training tasks included in those sessions in which the main objective, from a conditional perspective, is the overstimulation of strength and endurance capacities [17]. The TD and HID variables also differed in training sessions (4883 vs. 5437 m and 350 vs. 1027 m, respectively) from those reported for elite [16] and collegiate (TD = 2950 m) female soccer players [29]. This is probably because of variations in the competitive standards of the players [20], the training methodologies used [15], and the different high-intensity distance definition (16.0 vs 12.1 km·h$^{-1}$), which may have overestimated the training load because professional female players reach peak velocity > 24 km·h$^{-1}$ [30]. In addition, significant differences for TD, SPD, ACC, DCC, and PL were observed between MD-1 and the rest of the training sessions, confirming our hypothesis concerning the use of the tapering strategy within the microcycle and the existence of intra-microcycle differences.

Managing training loads may be facilitated if they are expressed as a percentage of official match-related loads, since this makes interpreting, communicating, and deciding the training prescription easier [31]. According to previous studies [15,28], our results showed that average team sprint distances in training sessions are lower compared to match days (23.5%, 64.4%, 24.7%, and 10.4% in MD-4, MD-3, MD-2, and MD-1, respectively). Therefore, matches constitute the main load of a training microcycle in female professional soccer players. In contrast to our results, Martín García et al. [15] showed that the highest sprint and peak velocities are reached in the MD-4 training session. These differences could be due to the number of training sessions that composed the microcycle (five vs. four). In all sessions, except MD-1, players reached velocities near the maximum velocity expressed in a match, but only FBs and FWs reached match velocity in MD-3. The exposure to HID in training sessions has been proposed as an injury-prevention strategy [32–34], so it is important for staff members to ensure that they design tasks to reach this intensity. On MD and MD-2, FWs showed many more sprints and longer sprint distances than CMs and CDs, because players in this position are expected to run into open space and break into the box to create goal-scoring opportunities, so it is necessary for them to make maximal sprint efforts [35].

In line with previous research [15,36], the SPD variable demonstrated the most variability in training sessions (107.1%, 31.5%, 28.8%, and 89.8% on MD-4, MD-3, MD-2, and MD-1, respectively), and on MD all variables showed lower variability (5.5%, 30.8%, and 6.5% for RD, HID, and SPD, respectively). These results could be explained by a combination of the inherent unpredictable nature of game-based training and the strategies coaches use to vary the stimulus for players to create training adaptations [15]. However, because the analysis of the variability in external load reported relevant values, the results should be interpreted with caution. In our study, the MD-4 and MD-1 training sessions presented the greatest variability (47.6% and 90.4% for HID and 107.1% and 89.8% for SPD), and MD had the least variability (30.8% and 6.5% for HID and SPD, respectively). Previous studies have reported greater variability (>80%) in HID and SPD values in the microcycle structure [15], or in training tasks (60–140%) [37] when compared to competitive matches (20–30%) [38,39]. Castillo et al., [17] obtained high between-session CVs (range = 10–100%) for distance covered at different intensities and short-term high-intensity actions across acquisition sessions in a Spanish professional team, with similar CV values (115%) obtained in English Premier League soccer players for high-intensity running during the in-season competitive phase [40]. Likewise, our results showed greater CV values for RD (7.1%), HID (38.2%), and SPD (107.1%), which coincides with previous literature [15,41]. This could be explained by contextual factors such as training task characteristics, physical fitness level, outcome, level of opposition, competitive level, and/ or tactical requirements [14,17], which can influence the players' external loads.

Despite only significant differences between microcycles in SPRD (m·min$^{-1}$) (M1 and M2 lower than M4) and HID (M3 lower than M2), M2 shows a ~ 30% greater than M3 and M4 and a ~ 28% and ~ 30% greater than M3 and M1 in HID and SPRD respectively. Previous studies have shown differences in microcycle load according to the time of the season [42] or the training objective [22]. A comprehensive analysis focused on playing positions could provide useful information for coaches to individualize the distribution of external training loads. During M4, FWs performed significantly more sprints (29.25±9.7 vs. 8.6±5.5, ~ 67%) and sprinted longer distances (624.0±140.5 vs. 165.7±117.2 m, ~75%) than CBs. This may be because the type of tasks implemented that week may have caused greater physical demands from FWs. In this sense, our results showed that FWs and FBs performed significantly more sprints than CDs and CMs on MD and MD-2 and FWs sprinted significantly longer distances than CMs on MD-2 without differences in the rest of the external load. These results are in line with those reported by Mohr et al. [9], who showed that during MD the attackers sprinted a greater distance (0.52±0.03 km vs. 0.33±0.05 km) but covered a similar (~10.2 km) total distance to defenders. The results further concur with previous findings whereby FWs covered the greatest distance at high intensity (consisting of HID and sprinting) [25]. However, other studies did not show differences between playing positions in TD and sprint distance [43]. Olivera et al. [28] These differences may be due to the different speed thresholds, as Vescovi and Favero [25] considered sprint distance covered to be at >20.0 km·h$^{-1}$ and Datson et al. [43] considered distance covered to be at >25.0 km·h$^{-1}$. Therefore, to provide references to make comparisons between studies, a methodological standardization of velocity thresholds is necessary to quantify external loads in female soccer players.

This study is not exempt from limitations, the main one being that all of the female players belonged to the same team, so external training load cannot be generalized to all clubs. In addition, the small sample size for each position and the playing formation this team used prevent generalizing to other female soccer players. A strength/speed session can cause increases in accelerations, but also decelerations and alter the load of the session. Therefore, quantifying these changes to make them uniform among players may be necessary. As the same external load can originate a different internal response, future studies that analyze these aspects together (internal and external load) may be of interest to understand the dose-response during the training process. However, the main strength of this study is the analysis of the inter- and intra-microcycle external loads across four consecutive weeks, reporting relevant data for coaches.

Quantifying external loads across training and matches could help coaches understand the dose-response across microcycles and, consequently, enhance the training periodization in female soccer. Because MD is the most demanding session of the week, adequate strategies for recovery in subsequent sessions are necessary in order for players to face the next game under optimal conditions. Likewise, considering the aforementioned differences found related to playing positions, special attention is required to periodize better training processes to allow players to prepare for match demands and reduce the risk of injury. In this sense, implementing soccer-specific speed drills is necessary for each playing position to increase external training load and prevent injuries.

## Conclusion

In summary, this study demonstrated that external load is similar for female professional soccer players during the inter-microcycle; however, the intra-microcycle revealed higher external loads in MD-3 in comparison to the rest of the training sessions. Likewise, MD was the most demanding session across the microcycle. In addition, considering the playing positions of

female players, FWs sprinted greater distances and performed a greater number of sprints than CBs in M4. Finally, differences were found in maximal velocity reached by FWs and FBs in comparison to CBs and CMs during the MD-2 training session and MD and between FWs and CMs in sprint distance during the MD-2 training session.

## Supporting information

**S1 Data.**
(XLSX)

## Author Contributions

**Conceptualization:** Elba Diaz-Seradilla, Alejandro Rodríguez-Fernández, José Antonio Rodríguez-Marroyo, Daniel Castillo, Javier Raya-González, José Gerardo Villa Vicente.

**Data curation:** Elba Diaz-Seradilla, Alejandro Rodríguez-Fernández, José Antonio Rodríguez-Marroyo, Daniel Castillo, Javier Raya-González, José Gerardo Villa Vicente.

**Formal analysis:** Elba Diaz-Seradilla, Alejandro Rodríguez-Fernández, José Antonio Rodríguez-Marroyo, Daniel Castillo, Javier Raya-González, José Gerardo Villa Vicente.

**Funding acquisition:** Elba Diaz-Seradilla, Alejandro Rodríguez-Fernández, José Antonio Rodríguez-Marroyo, Daniel Castillo, Javier Raya-González, José Gerardo Villa Vicente.

**Investigation:** Elba Diaz-Seradilla, Alejandro Rodríguez-Fernández, José Antonio Rodríguez-Marroyo, Daniel Castillo, Javier Raya-González, José Gerardo Villa Vicente.

**Methodology:** Elba Diaz-Seradilla, Alejandro Rodríguez-Fernández, José Antonio Rodríguez-Marroyo, Daniel Castillo, Javier Raya-González, José Gerardo Villa Vicente.

**Project administration:** Elba Diaz-Seradilla, Alejandro Rodríguez-Fernández, José Antonio Rodríguez-Marroyo, Daniel Castillo, Javier Raya-González, José Gerardo Villa Vicente.

**Resources:** Elba Diaz-Seradilla, Alejandro Rodríguez-Fernández, José Antonio Rodríguez-Marroyo, Daniel Castillo, Javier Raya-González, José Gerardo Villa Vicente.

**Software:** Elba Diaz-Seradilla, Alejandro Rodríguez-Fernández, José Antonio Rodríguez-Marroyo, Daniel Castillo, Javier Raya-González, José Gerardo Villa Vicente.

**Supervision:** Elba Diaz-Seradilla, Alejandro Rodríguez-Fernández, José Antonio Rodríguez-Marroyo, Daniel Castillo, Javier Raya-González, José Gerardo Villa Vicente.

**Validation:** Elba Diaz-Seradilla, Alejandro Rodríguez-Fernández, José Antonio Rodríguez-Marroyo, Daniel Castillo, Javier Raya-González, José Gerardo Villa Vicente.

**Visualization:** Elba Diaz-Seradilla, Alejandro Rodríguez-Fernández, José Antonio Rodríguez-Marroyo, Daniel Castillo, Javier Raya-González, José Gerardo Villa Vicente.

**Writing – original draft:** Elba Diaz-Seradilla, Alejandro Rodríguez-Fernández, José Antonio Rodríguez-Marroyo, Daniel Castillo, Javier Raya-González, José Gerardo Villa Vicente.

**Writing – review & editing:** Elba Diaz-Seradilla, Alejandro Rodríguez-Fernández, José Antonio Rodríguez-Marroyo, Daniel Castillo, Javier Raya-González, José Gerardo Villa Vicente.

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
