## [Decision Letter · Decision Letter 0]

20 Oct 2021

PONE-D-21-30806Inter- and intra-microcycle external load analysis in female professional soccer players: a playing position approach.PLOS ONE

Dear Dr. Rodríguez-Fernandez,

Thank you for submitting your manuscript to PLOS ONE. After careful consideration, we feel that it has merit but does not fully meet PLOS ONE’s publication criteria as it currently stands. Therefore, we invite you to submit a revised version of the manuscript that addresses the points raised during the review process.

The article is good and presents interesting findings. Minor changes should be considered following the reviewers recommendations.

We look forward to receiving your revised manuscript.

Kind regards,

Filipe Manuel Clemente, PhD

Academic Editor

PLOS ONE

Journal Requirements:

2.Please provide additional details regarding participant consent. In the ethics statement in the Methods and online submission information, please ensure that you have specified what type you obtained (for instance, written or verbal, and if verbal, how it was documented and witnessed). If your study included minors, state whether you obtained consent from parents or guardians. If the need for consent was waived by the ethics committee, please include this information.

Additional Editor Comments:

The article is good and presents interesting findings. Minor changes should be considered following the reviewers recommendations.

Reviewers' comments:

Reviewer's Responses to Questions

**Comments to the Author**

1. Is the manuscript technically sound, and do the data support the conclusions?

Reviewer #1: Yes

Reviewer #2: Yes

2. Has the statistical analysis been performed appropriately and rigorously? 

Reviewer #1: Yes

Reviewer #2: Yes

3. Have the authors made all data underlying the findings in their manuscript fully available?

Reviewer #1: Yes

Reviewer #2: Yes

4. Is the manuscript presented in an intelligible fashion and written in standard English?

Reviewer #1: Yes

Reviewer #2: Yes

5. Review Comments to the Author

Reviewer #1: Abstract

Ok

Introduction

Line 44: Include what is sprint speed

At a general level, a more exhaustive review of the importance of load control in football and its relationship with different performance parameters would be interesting.

Material and Methods

The previous acute / chronic burden accumulated before the study period was taken into account.

If competitive microcycle 1 is taken, the preseason phase previously existed. Did the players start with a similar load or was it different from the start?

The level of the rivals, were they similar, higher or lower than the level of the team studied?

What order was the session in? In the Strength, Endurance, Speed sessions, what kind of general work was done?

What kind of tactical tasks did they consist of? What% of time corresponds to each task in the session?

A Strength / Speed session can alter the parameters of decelerations or high intensity actions. Was it quantified in some way to make homogeneous efforts?

In games, how many minutes did the players play? Was it a homogeneous time? Were the exposure times in training sessions / matches relativized?

What were they based on to establish 15 or 20 min of warm-up? Was there a 17-minute warm-up? Those 5 minutes of margin that was included?

Statical

Ok

Results

Ok

Discussion

Were there any injuries during the study period either in training or in the game? Was the accumulated load on the players taken into account to regulate their intervention in training sessions?

It would be interesting to talk in the discussion about the% change in load with respect to the microcycles and also observe, the% load that each training session represents with respect to the previous match. It may be a future line of research.

high SPD values at -1 to be tapering phase

Since there are no standardized thresholds, why do the authors use <21 km? Is this speed based on men's soccer data? Has the speeds been relativized based on the individualization of each player? It may be that the players have their sprint level at 17-18-19 km and not at 21. Was it taken into account?

Reviewer #2: Inter- and intra-microcycle external load analysis in female professional soccer players:

a playing position approach

First of all, the reviewer would like to thank the authors for their work and efforts in trying to improve sports science knowledge.

General comments to the authors

Overall, this is a nice study that the study could have important practical information integrated with inter-and intra-microcycle external load according to playing position in female professional soccer players. The authors are commended on their efforts thus far. The study is well designed and well-written, with a great original article. However, I suggest only small corrections and the authors should update the recent references about inter-and intra-microcycle external load according to playing position in professional soccer players. These corrections and studies will allow improving the manuscript.

Abstract

Line 28: I think that we do not need this information (i.e., Primera Iberdrola League)

Instead of this, … who played to the same team in a regional league or something like that

Line 29: I think that we do not need this information in abstract (WIMU PRO, RealTrack Systems, Almería, Spain)

Line 35: Both fitness and conditioning experts ….

Introduction section

Line 47: As such, fitness and conditioning staff …

Line 62: the authors should add this article for professional male

Clemente, F., Silva, R., Arslan, E., Aquino, R., Castillo, D., & Mendes, B. The effects of congested fixture periods on distance-based workload indices: A full-season study in professional soccer players. Biology of Sport, 37(1), 37-44.

Line 62: the authors should add this article for professional female

Strauss, A., Sparks, M., & Pienaar, C. (2019). The use of GPS analysis to quantify the internal and external match demands of semi-elite level female soccer players during a tournament. Journal of sports science & medicine, 18(1), 73.

Materials and Methods section

This section is well designed and well-written.

Participants

Line 105: As the same as in abstract

Results section

This section is well designed and well-written.

Discussion section

Overall the discussion is well-written and incorporates relevant literature. However, the authors could use these recent articles in discussion, if you want

Oliveira, R., Martins, A., Nobari, H., Nalha, M., Mendes, B., Clemente, F. M., & Brito, J. P. (2021). In-season monotony, strain and acute/chronic workload of perceived exertion, global positioning system running based variables between player positions of a top elite soccer team. BMC Sports Science, Medicine and Rehabilitation, 13(1), 1-10.

Oliveira, R., Brito, J. P., Martins, A., Mendes, B., Marinho, D. A., Ferraz, R., & Marques, M. C. (2019). In-season internal and external training load quantification of an elite European soccer team. PloS one, 14(4), e0209393.

Clemente, F. M., Silva, R., Ramirez-Campillo, R., Afonso, J., Mendes, B., & Chen, Y. S. (2020). Accelerometry-based variables in professional soccer players: comparisons between periods of the season and playing positions. Biology of Sport, 37(4), 389.

Hasan, U. C., Silva, R., & Clemente, F. (2021). Weekly variations of biomechanical load variables in professional soccer players: Comparisons between playing positions. Human Movement, 22(1), 19-34.

Silva, R., Ceylan, H. I., Badicu, G., Nobari, H., Carvalho, S. A., Sant’Ana, T., ... & Clemente, F. M. (2021). Match-to-match variations in external load measures during congested weeks in professional male soccer players. J. Men’s Health, 1-11.

Stevens, T. G., de Ruiter, C. J., Twisk, J. W., Savelsbergh, G. J., & Beek, P. J. (2017). Quantification of in-season training load relative to match load in professional Dutch Eredivisie football players. Science and Medicine in Football, 1(2), 117-125.

Figure and Tables

This section is well designed and well-written.

6. PLOS authors have the option to publish the peer review history of their article (what does this mean?). If published, this will include your full peer review and any attached files.

Reviewer #1: No

Reviewer #2: No

---

## [Author Response · Author response to Decision Letter 0]

19 Nov 2021

Response to Reviewers

Response to reviewer 1:

First of all, we would like to express our gratitude to Reviewer 1 for the time in reviewing our paper and for providing us comments/suggestions helpful to improve this paper quality. We are very proud of your comments. We think that your observations have improved the manuscript. We have answered point-to-point in yellow in this document and in the new version of the manuscript.

Reviewer #1: Abstract

Ok

Introduction

Line 44: Include what is sprint speed

Response: Dear reviewer, we have included values of sprint speed.

At a general level, a more exhaustive review of the importance of load control in football and its relationship with different performance parameters would be interesting.

Response: Dear reviewer, we have modified the introduction section including a more exhaustive review of the importance of load control. You can see in introduction section“ Inappropriately training loads can cause overuse injuries [18] and increased injury risk is associated with spikes in workload (i.e., overloading) and low chronic workloads (i.e., underloading) [19]. Given that quantification of the workload has become an essential aspect in soccer performance [11].”

[11] Malone J, di Michele R, Morgans R, et al. Seasonal training-load quantification in elite English Premier League soccer players. International Journal of Sports Physiology and Performance 2015; 10: 489–497.

[18] Drew MK, Finch CF. The Relationship Between Training Load and Injury, Illness and Soreness: A Systematic and Literature Review. Sports medicine (Auckland, NZ) 2016; 46: 861–883.

[19] Hulin BT, Gabbett TJ, Lawson DW, et al. The acute: Chronic workload ratio predicts injury: High chronic workload may decrease injury risk in elite rugby league players. British Journal of Sports Medicine 2016; 50: 231–236.

Material and Methods

The previous acute / chronic burden accumulated before the study period was taken into account.

Response: Dear reviewer, we have considered previous workload. In this sense the previous microcycle to the experimental design was a recovery microcycle and all the following microcycles were performance typology in season (after preseason finished). See the next response.

If competitive microcycle 1 is taken, the preseason phase previously existed. Did the players start with a similar load or was it different from the start?

Response: Thanks for your appreciation. We include in methods section an explication about your concern: “Microcycle 1 was performed in the middle of the first part of the season, 4-weeks after the end of the pre-season with a similar workload in all players”

The level of the rivals, were they similar, higher or lower than the level of the team studied?

Response: Dear reviewer, we know the importance of the team level as a contextual factor in the match demands. It is difficult to predict the level of an opponentin early phases of the season, but in this case the four microcycles were placed against opponents with similar level according to their final classification during the previous season.

What order was the session in? In the Strength, Endurance, Speed sessions, what kind of general work was done?

Response: In Procedures section, we agree more exhaustive explanation: “The usual distribution of the microcycle was as follows: MD-4: recovery session and general resistance; MD-3: resistance and endurance session (i.e., plyometric drills, strength stations and game simulation); MD-2: speed and tactical approximation (i.e., sprint running and large small sided games); and MD-1: activation session (reaction speed and small sided games), targeting the three main physical capacities and similar to that shown by previous studies with male soccer players (Buchheit et al., 2018; Castillo et al., 2019). In MD-3 tactical drills (i.e., medium and large small games with tactical constrains) was performed bay female soccer players”.

Buchheit, M., Lacome, M., Cholley, Y., & Simpson, B. M. (2018). Neuromuscular responses to conditioned soccer sessions assessed via GPS-Embedded accelerometers: Insights into tactical periodization. International Journal of Sports Physiology and Performance, 13(5), 577–583. https://doi.org/10.1123/ijspp.2017-0045

Castillo, D., Raya-González, J., Weston, M., & Yanci, J. (2019). Distribution of External Load During Acquisition Training Sessions and Match Play of a Professional Soccer Team. Journal of Strength and Conditioning Research. https://doi.org/10.1519/jsc.0000000000003363

What kind of tactical tasks did they consist of? What% of time corresponds to each task in the session?

Response: We agree with the reviewer, so a better explanation of the characteristics of the sessions have been included in the procedures section: “Tactical task (i.e., conditioned games in medium and large spaces simulating competition situations, superiorities and inferiorities) represents 20%, 40% and 40% in MD-4, MD-3 and MD-2 respectively”

A Strength / Speed session can alter the parameters of decelerations or high intensity actions. Was it quantified in some way to make homogeneous efforts?

Response: thank you for your comment. In this sense we no quantified the decelerations or high intensity actions in aim to make homogeneous efforts. We include how possible limitation: Line 302: A strength / speed session can cause increases in accelerations, but also decelerations and alter the load of the session. Therefore, quantifying these changes to make them uniform among players may be necessary.

In games, how many minutes did the players play? Was it a homogeneous time? Were the exposure times in training sessions / matches relativized?

Response: Dear reviewer, only player who that completed at least 60 min were analyzed. We have included it on procedures section. “The external load of the microcycle was analyzed only in those players who played ≥ 60 min in MD”

What were they based on to establish 15 or 20 min of warm-up? Was there a 17-minute warm-up? Those 5 minutes of margin that was included?

Response: To determine warm up time we used usual time in female soccer players (Isla et al., 2021, Pardos et al., 2019). In addition, it is the usual time that the coach used in the warm-up routines in this team and in order not to limit him there was a margin of 5 minutes. Following references support this concern:.

Isla, E., Romero-Moraleda, B., Moya, J. M., Esparza-Ros, F., & Mallo, J. (2021). Effects of a Neuromuscular Warm-Up Program in Youth Female Soccer Players. Journal of human kinetics, 79, 29–40. https://doi.org/10.2478/hukin-2021-0080

Pardos-Mainer, E., Casajús, J. A., & Gonzalo-Skok, O. (2019). Adolescent female soccer players' soccer-specific warm-up effects on performance and inter-limb asymmetries. Biology of sport, 36(3), 199–207. https://doi.org/10.5114/biolsport.2019.85453

Statical

Ok

Results

Ok

Response: Thanks for the consideration. However, we have added one column in each table showing the absolute value of high intensity distance covered HID (m) by female soccer player, so the HID and distance to sprint (SPD) covered are shown in absolute values (m) and relative to time (m·min-1) in each table.

Discussion

Were there any injuries during the study period either in training or in the game? Was the accumulated load on the players taken into account to regulate their intervention in training sessions?

Response: Thank you for your comment. In this sense, no injuries were occurred during the analysis period (4 microcycles). Acute:Chronic load was not taken into account since the authors were not part of the team staff. Today women's soccer, despite analyzing a first division team (i.e., Primera Iberdrola League), does not have the same resources as men's soccer.

It would be interesting to talk in the discussion about the% change in load with respect to the microcycles and also observe, the% load that each training session represents with respect to the previous match. It may be a future line of research.

Response: Dear reviewer, we agree that It would be interesting to talk about the % change in load in relation to the microcycles and also to observe the % load that each training session represents with respect to the previous match. Although, it is not directly related to the objectives of our study, we include values in the discussion section. Line 291-295 Thanks for the contribution, it is an interesting future line of research.

high SPD values at -1 to be tapering phase

Response: we agree but in MD-1 CV is 89.8%, 59.1% and 88.2% to SPD in m, m·min-1 and nº respectively.

Since there are no standardized thresholds, why do the authors use <21 km? Is this speed based on men's soccer data? Has the speeds been relativized based on the individualization of each player? It may be that the players have their sprint level at 17-18-19 km and not at 21. Was it taken into account?

Dear reviewer, indeed different speed thresholds have been proposed for female soccer players, some of them related to male values or proposing some lower ones. In our case we have taken into account: Physical Analysis of the FIFA Women’s World Cup France 2019TM (https://digitalhub.fifa.com/m/4f40a98140d305e2/original/zijqly4oednqa5gffgaz-pdf.pdf)

Reviewer #2: Inter- and intra-microcycle external load analysis in female professional soccer players:

a playing position approach

First of all, the reviewer would like to thank the authors for their work and efforts in trying to improve sports science knowledge.

General comments to the authors

Overall, this is a nice study that the study could have important practical information integrated with inter-and intra-microcycle external load according to playing position in female professional soccer players. The authors are commended on their efforts thus far. The study is well designed and well-written, with a great original article. However, I suggest only small corrections and the authors should update the recent references about inter-and intra-microcycle external load according to playing position in professional soccer players. These corrections and studies will allow improving the manuscript.

Response: First of all, we would like to express our gratitude to Reviewer 2 for the time in reviewing our paper and for providing us comments/suggestions helpful to improve this paper quality. We are very proud for your comments which help us to improve the manuscript substantially. We have answered point-to-point in highlight (yellow) in this document and in the new version of the manuscript.

Abstract

Line 28: I think that we do not need this information (i.e., Primera Iberdrola League).

Instead of this, … who played to the same team in a regional league or something like that

Response: We agree with you. Corrected: “… who belonged to the same team in first national division”

Line 29: I think that we do not need this information in abstract (WIMU PRO, RealTrack Systems, Almería, Spain)

Response: We agree. Deleted.

Line 35: Both fitness and conditioning experts ….

Response: Dear reviewer, corrected.

Introduction section

Line 47: As such, fitness and conditioning staff …

Response: Corrected.

Line 62: the authors should add this article for professional male

Clemente, F., Silva, R., Arslan, E., Aquino, R., Castillo, D., & Mendes, B. The effects of congested fixture periods on distance-based workload indices: A full-season study in professional soccer players. Biology of Sport, 37(1), 37-44.

Line 62: the authors should add this article for professional female

Strauss, A., Sparks, M., & Pienaar, C. (2019). The use of GPS analysis to quantify the internal and external match demands of semi-elite level female soccer players during a tournament. Journal of sports science & medicine, 18(1), 73.

Response: Dear reviewer, thanks for the specific reference proposed: Include

Materials and Methods section

This section is well designed and well-written.

Participants

Line 105: As the same as in abstract

Response: corrected.

Results section

This section is well designed and well-written.

Response: Thanks for the consideration. However, we have added one column in each table, showing the absolute value of high intensity distance covered HID (m) by female soccer player, so the HID and distance to sprint (SPD) covered are shown in absolute values (m) and relative to time (m·min-1) in each table.

Discussion section

Overall the discussion is well-written and incorporates relevant literature. However, the authors could use these recent articles in discussion, if you want

Oliveira, R., Martins, A., Nobari, H., Nalha, M., Mendes, B., Clemente, F. M., & Brito, J. P. (2021). In-season monotony, strain and acute/chronic workload of perceived exertion, global positioning system running based variables between player positions of a top elite soccer team. BMC Sports Science, Medicine and Rehabilitation, 13(1), 1-10.

Oliveira, R., Brito, J. P., Martins, A., Mendes, B., Marinho, D. A., Ferraz, R., & Marques, M. C. (2019). In-season internal and external training load quantification of an elite European soccer team. PloS one, 14(4), e0209393.

Clemente, F. M., Silva, R., Ramirez-Campillo, R., Afonso, J., Mendes, B., & Chen, Y. S. (2020). Accelerometry-based variables in professional soccer players: comparisons between periods of the season and playing positions. Biology of Sport, 37(4), 389.

Hasan, U. C., Silva, R., & Clemente, F. (2021). Weekly variations of biomechanical load variables in professional soccer players: Comparisons between playing positions. Human Movement, 22(1), 19-34.

Silva, R., Ceylan, H. I., Badicu, G., Nobari, H., Carvalho, S. A., Sant’Ana, T., ... & Clemente, F. M. (2021). Match-to-match variations in external load measures during congested weeks in professional male soccer players. J. Men’s Health, 1-11.

Stevens, T. G., de Ruiter, C. J., Twisk, J. W., Savelsbergh, G. J., & Beek, P. J. (2017). Quantification of in-season training load relative to match load in professional Dutch Eredivisie football players. Science and Medicine in Football, 1(2), 117-125.

Response: Dear reviewer, thanks for the specific and actual reference suggestion. We have included those related to the sections discussed in the discussion section. 

Figure and Tables

This section is well designed and well-written.

Response: Thanks for the consideration.

6. PLOS authors have the option to publish the peer review history of their article (what does this mean?). If published, this will include your full peer review and any attached files.

Do you want your identity to be public for this peer review? For information about this choice, including consent withdrawal, please see our Privacy Policy.

Reviewer #1: No

Reviewer #2: No

---

## [Decision Letter · Decision Letter 1]

22 Feb 2022

Inter- and intra-microcycle external load analysis in female professional soccer players: a playing position approach.

PONE-D-21-30806R1

Dear Dr. Rodríguez-Fernandez,

We’re pleased to inform you that your manuscript has been judged scientifically suitable for publication and will be formally accepted for publication once it meets all outstanding technical requirements.

Kind regards,

Filipe Manuel Clemente, PhD

Academic Editor

PLOS ONE

Additional Editor Comments (optional):

Reviewers' comments:

Reviewer's Responses to Questions

**Comments to the Author**

1. If the authors have adequately addressed your comments raised in a previous round of review and you feel that this manuscript is now acceptable for publication, you may indicate that here to bypass the “Comments to the Author” section, enter your conflict of interest statement in the “Confidential to Editor” section, and submit your "Accept" recommendation.

Reviewer #1: All comments have been addressed

Reviewer #2: All comments have been addressed

2. Is the manuscript technically sound, and do the data support the conclusions?

Reviewer #1: Yes

Reviewer #2: Yes

3. Has the statistical analysis been performed appropriately and rigorously? 

Reviewer #1: Yes

Reviewer #2: Yes

4. Have the authors made all data underlying the findings in their manuscript fully available?

Reviewer #1: Yes

Reviewer #2: Yes

5. Is the manuscript presented in an intelligible fashion and written in standard English?

Reviewer #1: Yes

Reviewer #2: Yes

6. Review Comments to the Author

Reviewer #1: (No Response)

Reviewer #2: Overall, this is a nice study that the study could have important practical information integrated with interand intra-microcycle external load according to playing position in female professional soccer players. The

authors are commended on their efforts thus far. Accepted

7. PLOS authors have the option to publish the peer review history of their article (what does this mean?). If published, this will include your full peer review and any attached files.

Reviewer #1: No

Reviewer #2: No

---

## [Editor Report · Acceptance letter]

11 Mar 2022

PONE-D-21-30806R1 

Inter- and intra-microcycle external load analysis in female professional soccer players: a playing position approach. 

Dear Dr. Rodríguez-Fernández:

I'm pleased to inform you that your manuscript has been deemed suitable for publication in PLOS ONE. Congratulations! Your manuscript is now with our production department. 

Kind regards, 

on behalf of

Dr. Filipe Manuel Clemente 

Academic Editor

PLOS ONE